# Modeling of Forest Communities' Spatial Structure at the Regional Level through Remote Sensing and Field Sampling: Constraints and Solutions

**Ivan Kotlov [1],\* and Tatiana Chernenkova [2]**

[1] Severtsov Institute of Ecology and Evolution, Russian Academy of Sciences, Leninsky Ave. 33, 119071 Moscow, Russia

[2] Institute of Geography, Russian Academy of Sciences, Staromonetniy Pereulok 29, 119017 Moscow, Russia; chernenkova50@mail.ru

\* Correspondence: ikotlov@gmail.com; Tel.: +7-903-973-83-10

**Abstract:** This study tests modern approaches to spatial modeling of forest communities at the regional level based on a supervised classification. The study is conducted by the example of mapping the composition of forest communities in a large urbanized region (the Moscow Region, area 4.69 million hectares). A database of 1684 field descriptions is used as sample plots. As environmental variables, Landsat spectral reflectances, vegetation indices (5 images), digital elevation model and morphometric parameters of the relief, 54 layers in total, are used. Additionally, the Palsar-2 radar dataset is included. The main mapped units are formations and groups of associations identified on the basis of the ecological-phytocoenotic classification. Formations and groups of associations are similar in semantics and principles of allocation to units of forest typology. It is shown that the maximum entropy method has a wide range of applications, in particular, for mapping the typological diversity of forest cover. The method is used in combination with geographically structured spatial jack-knifing, spatial rarefication of occurrence data and independent testing of model feature classes and regularization parameters. Spatial rarefication is a critical technique when points are not evenly distributed in space. The resulting model of the spatial structure of forest cover is based on the integration of the best models of each thematic class of different types of forest cover into a single cartographic layer. It is shown that under conditions of uneven and sparse distribution of points, it is possible to provide an average point matching level of 0.45 for formations and 0.29 for association groups. Herewith, the spatial structure and the ratio of the formation's composition correspond to the official data of the forest inventory. An attempt is made to identify and evaluate the distribution of more detailed syntaxonomic units: association groups. The necessary requirements for improving the quality of the forest cover model of the study area for 2 hierarchical typological units of forest cover are formulated. These include the additional sampling in order to equalize their spatial density, as well as to achieve equality of samples based on stratification according to the resulting map.

**Keywords:** spatial modeling; forest formation; association group; ecological-phytocoenotic classification; MaxEnt; SDMtoolbox; spatial modeling; Moscow Region; Landsat

## 1. Introduction

The mapping of the spatial structure of forest communities is an integral part of biodiversity research and environmental planning at the regional level. There are three basic constraints on forest data: First, the data on forest spatial structure must be up-to-date and be able to be regularly updated with a step of 2–5 years. Second, the data should equally describe the parameters of biodiversity and

species composition of forest communities, and not just stocks of industrial wood species. Third, the combination of local measurement data in the process of ground-based research with multispectral satellite imagery data and quantitative methods of their processing should ensure the display of important information about the structure and properties of vegetation on the map. The need to obtain more diverse and detailed information in the forest inventory is formulated as a result of the activities of international programs that regulate certain actions not only in the environmental, but also in the social and economic spheres [1].

In many countries abroad, the National Forest Inventories (NFIs) system is based on the nature of remote information combined with ground data laid down in a regular network of permanent sample plots [2]. A number of requirements are imposed on modern mapping of natural objects based on supervised classification [3]: sampling design [4], preliminary stratification of the study area into homogeneous strata [5,6], random uniform distribution of sampling points within strata and equality of samples between strata. In particular, this is necessary to reduce spatial autocorrelation within field data samples [7,8]. Another important requirement is the correspondence of the sample size to the minimum value, that varies according to different studies from 25 to 80 sample elements for each modeled object [9]. These requirements are often difficult to meet due to the fact that long-term field data collection programs that were carried out 5–10 years ago did not take many of these factors into account [10]. Due to the limited capabilities, field materials do not possess such properties a priori (in whole or in part); therefore, appropriate preliminary preparation of samples is required.

In Russia, unlike NFIs, the location of sample plots is mainly irregularly distributed, their density per area is at least 6 times less, and the spatial distribution has strong bias to the road network and settlements [11]. Moreover, the data of the state forest inventory are either officially classified or available as old paper maps. Under these conditions, the collection of scientific data on the state of plant communities is carried out by individual scientific institutes or teams extremely rarely on a systematic basis and is characterized by a number of shortcomings: (1) Uncertainty of determination of forest association groups by different researchers and (2) uneven distribution of field data in space due to transport infrastructure and inaccessibility of territories. In addition, horizontal uncertainty of GPS L1 receivers (Level 1 for civilian use) under dense forest canopy makes a negative contribution [12]. The reasonings above lodge a challenge of searching the most effective approaches and methods for modeling and mapping the spatial structure of forest communities using the available sources of data.

The Moscow Region is selected as a test area. Taking into account the strengthening of urban planning activities, the development of country and cottage construction and recreation in the region, the maintenance of ecological and social functions of the "green belt" forests is extremely important [13–15]. Concern about the state of forest plantations is noted not only on the part of NGOs (non-governmental organizations) and experts [16], but also at the state and regional level [17].

To date, there are three large-scale sources of information on the composition and spatial structure of forests in the Moscow Region: (1) materials of the state forest inventory, performed in 1995–2000 [17], (2) map of the vegetation cover of the Moscow Region, made by the team of authors at Moscow University in 1996 [18] and (3) map of terrestrial ecosystems of the Moscow Region [19]. The first two sources are characterized by significant prescription. In addition, the state inventory is based on the collection of information on the stocks of industrial wood species and, to a much lesser extent, on the data on the composition of the ground layers. It also should be noted that according to Russian forest management guidelines, the systematic error is allowed for forest inventory data. This error may reach 10–20% of species proportion in 32% of controlled forest patches [20]. The third source contains only 6 generalized forest types, which is not enough for a biodiversity inventory. Thus, for the Moscow Region (and probably for most regions of Russia), there is an obvious lack of up-to-date cartographic material on the spatial structure of vegetation, primarily, forest cover, which raises the question of the availability of reliable field data on the state of the vegetation (forest) cover.

A variety of modeling approaches are available currently [21]. The Linear Discriminant Analysis was tested by the authors and it was demonstrated that non-linear features of environmental variables

might potentially improve the robustness of the model [22]. In the current study, the SDMtoolbox (Spatial Distribution Modeling toolbox 2.4, Durham, NC/Manhattan, NY/Carbondale, IL, USA) is chosen as a modeling tool. The SDMtoolbox is a python-based ArcGIS 10.7 toolbox (Redlands, CA, USA) for spatial studies of ecology, evolution and genetics. The SMDtoolbox is chosen because it includes the basic MaxEnt (Maximum Enthropy, Manhattan, NY, USA) algorithm and a number of additional tools necessary to control the autocorrelation of spatial data [23]. Among other methods, MaxEnt is shown as an effective tool for non-linear interactions between response and predictor variables, and is robust to small sample sizes [21]. MaxEnt is used not only for the Species Distribution Model, but also for a wide range of natural phenomena, for instance, for tree pests monitoring (Salento Peninsula, Italy) [24], to create predictive risk maps for soil-transmitted helminth infections in Thailand [25], to study invasive species (Korea) [26] and critically endangered Alaotran gentle lemur (Madagascar) [27]. MaxEnt is even applied in mineral prospectivity analysis (Nanling, China) [28] or as the model for the prediction of landslide patterns (Arno Basin, Italy) [29]. The geology, soil, climate and vegetation spatial data are used in the above-mentioned cases along with remote sensing data. It can be assumed that the maximum entropy method together with the MaxEnt software is the universal geographical tool for spatial modeling and mapping.

It is also a developing practice to apply spatial modeling to such natural phenomena as forest formations. For example, the habitat suitability modeling was performed for 10 types of forest formations in Europe. The Random Forest model based on 1 km climate environmental variables and more than 6000 field data forest inventory plots were used in that study. The overall accuracy of the final map was 76% [30]. Finite mixture model was applied to a national forest inventory of Italy consisting of 6714 plots with a measure of abundance for 27 tree species, and the map of potential forest types was produced also based on 1 km climate data supplemented with some geological and soil data [31]. According to the study of Panamanian tree species, their distribution appears to be primarily determined by dispersal limitation, then by environmental heterogeneity. This study used a permutation-based regression model computed on distance matrices and a hierarchical clustering of the tree composition to construct a predictive map of forest types of the Panama Canal. Fifty-three sample plots describing the floristic composition along with climatic data, elevation, geologic formation and slope are used in the referred study [32]. The study of southern Atlantic Rainforest formations (Brazil) aimed to verify the existence of indicator species and identify relationships among distributions of tree species with environmental and spatial variables. The study was based on 21 sample plots and altitude and climatic variables [33]. The forest formations are often referred to in the aforementioned studies as Floristic Patterns or Species Composition Patterns. The satisfactory results are shown for application of MaxEnt in land-cover classification and land change analysis. For example, the study in Trentino-South Tyrol, Italy, developed land cover and difference maps between 1976 and 2001 based on multispectral data and topographic variables. MaxEnt applied to land cover classes can provide reliable data, especially when referring to classes with homogeneous texture properties and surface reflectance [34].

The purpose of this study is to assess the utility of modern approaches in spatial modeling of forest communities for the example of the Moscow Region, based on field data obtained outside the state forest inventory. The tasks of this study are dictated by the need to develop and adapt optimal methods for managing the array of field descriptions unevenly distributed in space and between syntaxonomic units, as well as to develop a probabilistic cartographic model of forest communities at the regional level, as an alternative to the generalized official data of the forest inventory.

## 2. Materials and Methods

### 2.1. Study Area

The Moscow Region is located in the central part of the East European (Russian) Plain: 35°10′–40°15′ East, 54°12′–56°55′ North, and covers an area of 4.69 million hectares (including Moscow:

0.26 million hectares) (Figure 1). The population is 20.4 million people (Moscow: 12.7 million people). The average population density is 2.29 people per square kilometer. The region has several important natural and phytogeographical boundaries. The significant border is the south edge of the natural range of the spruce forests in the broad-leaved spruce forests zone.

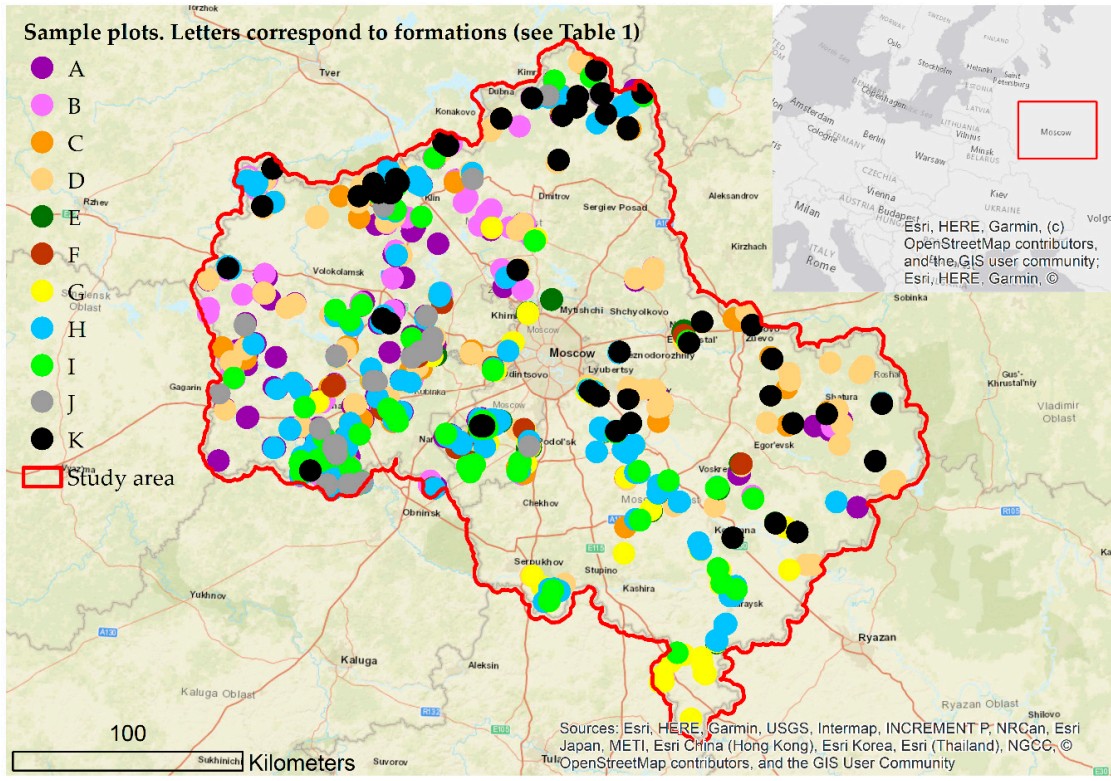

**Figure 1.** Study area and field sample plots. The locations of 1684 sample plots are shown on the figure along with the formation identity of each sample plot (color). A: Spruce, B: Spruce—aspen/birch, C: Pine—spruce, D: Pine, E: Oak—Spruce, F: Broadleaf—spruce, G: Linden, H: Birch, I: Aspen, J: Grey alder, K: Black alder.

Due to the high soil fertility, the forest cover of the region has experienced an extensive anthropogenic impact (felling, plowing) for several centuries. At the beginning of the 20th century, and especially after the Second World War, there was a significant change in the direction of impact: the use of forestry practices on the site of the former arable land. In 1947, all forests of the Moscow Region were recognized as a green zone, including a ban on industrial felling. However, up to the end of the 20th century, there was an increase in the pace of industrial development: the construction and operation of machine-building plants and related infrastructure, including enterprises of the energy complexes and oil refining complexes [35]. This has led to an increase in emissions of pollutants into the atmosphere and hydrosphere. At the beginning of the 21st century, the direction of influence changed to post-industrial. The growth of the population along with the construction of housing stock and transport infrastructure started in connection with the development of the financial sector of the economy [36,37]. In addition, deterioration in the volume of forestry activities aimed at maintaining the sustainability of forest plantations was recorded. The nature protection regime violations are as follows: unauthorized felling and household/industrial waste dumps. These violations result in large pest outbreaks, forest fires and degradation of species composition [17].

*2.2. Design of the Study*

1684 sample plots (including 1494 forest sample plots) were collected during 2010–2019 (Figure 1). The sample plots' locations selection is based on the principle of representativeness for the main

species prevailing in the forest stand, forest types, taking into account age groups, as well as the ecology of the habitat. The sampled vegetation communities are homogeneous in terms of the general floristic composition, the composition of the dominants of each layer, physiognomy (aspect, community structure) and habitat conditions. According to the methodology, the sample plots were limited to $20 \times 20$ m and located at a distance of at least 200 m from each other. When localizing the sample plots, the representation from the main types of communities was taken into account. Lack of road infrastructure was a serious limitation in the uniform laying of test plots. The following properties of vegetation communities are recorded at each sample plot:

- Composition and structure of the tree layer (projective crown cover, average height of mature trees and undergrowth).
- Complete species composition of shrub, grass-dwarf shrub and moss layers, with an estimate of the cover in percent.
- Species saturation of the plants of the ground layers, estimated as the average number of species per unit area (to assess the species diversity).

The previous study of Chernenkova [38] has shown that the ecological-phytocenotic classification is best for analysis of communities in this region. The techniques allows for great differentiation of communities across the region [38]. The use of units of ecological-phytocenotic classification in the rank of formation and association groups is explained by a number of reasons: (1) good correspondence of typological and mapped units, (2) compliance with Russian units of forest typology and (3) consideration of rare types of forest areas, as well as secondary derivative communities, which is important from an environmental point of view.

Formations are identified according to the dominant forest species. Each formation is represented by communities with a different combination of common dominants in the lower stories. The classifications of syntaxons at the level of association groups is carried out according to the dominant ecological and morphological groups of plants of subordinate stories [38]. The association groups are connected with the features of the herb-dwarf shrub layer, trophic conditions and moisture conditions. Comparison of the formations' composition and association groups allows one to assess the direction of successional development, the degree of anthropogenic impact and the stability of forest communities.

For mapping the spatial structure, the maximum entropy is chosen [39]. The choice of method is based on the nature of the field data collection. There are two main ways to collect data on the distribution of natural objects [40], which require the use of various modeling algorithms: One is collection of only occurrence points. With this method of analysis (it is called "presence only" or "presence/background" presence analysis), one can use GPS data, materials of collections and herbaria, publications describing the places of species registration. Another is separate collection of occurrence and absence points. This is called a "presence absence" analysis. The possibility of collecting correct information about the absence of a natural phenomenon in any territory is debatable and may turn out to be false. The maximum entropy method is designed specifically for processing "presence/background" data [9].

SDMtoolbox software includes additional tools, namely calibration of multiple models, their testing and final selection of the best model using the quality indicators [41]. An important part of the tool is the spatial jack-knifing (or geographically structured k-fold cross-validation) [42]. This tool tests evaluation performance of spatially segregated, spatially independent localities. The tool also allows for testing different combinations of model feature class types (linear, quadratic, hinge, product and threshold) and regularization multipliers to optimize MaxEnt model performance.

Landsat 8 and Landsat 5 spectral reflectances and spectral indices as well as Shuttle Radar Topography Mission (SRTM) digital elevation model (DEM) and morphometric variables are used as the environmental variables [43,44]. Palsar-2 25 m resolution images are also included: ortho and slope-corrected backscattering coefficient (horizontal–horizontal (HH) and horizontal–vertical (HV) polarization) for 2019 [45]. To manage the autocorrelation, the variables that have 95% correlation

with other variables are removed, and 54 of 83 variables are left, the list of variables is provided in Appendix A Table A1. The Global Forest Watch dataset is utilized to prepare forest/non-forest masks as well as the loss year mask and water mask [46].

The modeling of forest cover is performed within two hierarchical levels: (1) forest formations and (2) association groups. Forest formations are aggregated syntaxons that are warranted by statistically sufficient and homogenous training samples. This makes forest formations more robust for spatial modeling. Association groups are more divisional, in that they have heterogenous training samples and thus they are more sensitive natural objects for modeling. The overall quality of modeling is evaluated by two confusion matrices.

On the first upper level, forest formations are modeled. Using the Global Forest Watch dataset, the territory is divided into two strata based on 30% forest cover threshold and forest loss/gain data: forest stratum and non-forest stratum. Raster masks are used to apply strata during mapping. Multiple approaches are tested for selection of bias layers [40–42]: no bias, and 10 and 25 km biases around occurrence points. The best results (highest area under the curve and 1-omission error) are obtained using forest layer as bias for forest formations and non-forest layer as bias for open lands and agriculture.

The systematic sampling approach that showed the best results along with bias layer, clustering, splitting and background restriction approaches is utilized [21]. To provide balance between sample sizes and sample equality, the sample size for each formation is systematically decimated to around 100 sample plots, because MaxEnt allows to include all non-linear feature class types only when the size is over 80 samples. To achieve 100 sample plots as well as to equalize sample sizes of formations and to reduce spatial autocorrelation, the following compromising approach is used. The spatial rarefication of occurrence data is applied.

Spatial jack-knifing is performed by three regions, five model feature class types are used (linear, quadratic, hinge, product and threshold) and three regularization multipliers (0.5, 1, 2). Final models of formations are integrated into one map by the method of highest position in ArcGIS. The confusion matrix is calculated between the final map and the initial sample plots.

On the second level, the modeling of association groups is performed. The number of sampling points varies significantly from 9 to 154: average 48.96, median 35, standard deviation 39.95. For this reason, no transformation and no spatial rarefication of the number of sample plots is applied. For every association group, the formation mask is applied. The confusion matrix is also calculated.

## 3. Results

### 3.1. Pre-Processing of Samples

Table 1 shows the hierarchical structure of formations (in columns) and association groups (in rows) of the Moscow Region, in accordance with the previously published results of the ecological-phytocenotic classification [43]. The cells indicate the number of field sample plots for the association groups.

Formation A: The composition of spruce forests is complex (combinations of spruce with birch, aspen, pine and broad-leaved species) and it is similar to the composition of the zonal primary coniferous broad-leaved forests. The proportion of silviculture is high (mainly monodominant spruce forests) [44]. The species composition of the subordinate stories (grass-dwarf shrub and moss stories) is represented by the full spectrum of transitions from boreal to nemoral types. A relatively small number of community types is noted in the composition of boreal spruce forests (small herb and small herb-green moss groups). While, subnemoral (small herb-broad herb) and nemoral (broad herd) groups of spruce forests have a higher coenotic diversity, due to the higher participation of other tree species, and the variety of combinations of dominant land cover species.

**Table 1.** Ecological-phytocenotic classification of forest communities and the number of points of association groups before and after spatial rarefication.

| Formations [1] | Spruce | Spruce-Aspen/Birch | Pine-Spruce | Pine | Oak-Spruce | Broad Leaf-Spruce | Linden | Birch | Aspen | Grey Alder | Black Alder |
|---|---|---|---|---|---|---|---|---|---|---|---|
| Association groups | A | B | C | D | E | F | G | H | I | J | K |
| DShG | 37 | 30 | 32 | 46 | | | | | | | |
| Sh | 39 | 22 | 16 | 23 | | | | 9 | | | |
| ShBh | 146 | 78 | 44 | 35 | | | | 29 | | | |
| Bh | 147 | 102 | 41 | 64 | 57 | 38 | 112 | 154 | 84 | | |
| MhBh | | | | | | | | 18 | 16 | 30 | 24 |
| Gm | | | | | | | | 17 | | | 31 |
| H | | | | 15 | | | | 24 | | | |
| DHS | | | | 46 | | | | 10 | | | |
| Total number of sample plots | 369 | 232 | 133 | 229 | 57 | 38 | 112 | 261 | 100 | 30 | 55 |
| Spatial rarefication, km | 10 | 10 | 1 | 5 | - | - | - | 10 | - | - | - |
| Number of sample plots after rarefication | 97 | 87 | 93 | 82 | | 95 [2] | 112 | 95 | 100 | 85 [3] | |

[1] DShG: dwarf shrubs-small herb-green moss, Sh: small herb, ShBh: small herb-broad herb, Bh: broad herb, MhBh: moist herb-broad herb, Gm: grass-marsh, H: herb, DHS: dwarf shrubs-herb-sphagnum. [2] Merged formations: oak and broadleaf (E and F). [3] Merged formations: gray alder and black alder (J and K).

Formation B: spruce–aspen/birch forests. This group of formations includes communities where spruce and birch are represented in equal proportions (with a small admixture of aspen). Spruce-small-leaved forests are interpreted by many researchers [47,48] as a short-term stage of spruce forests that form at the felling site as a result of both spontaneous succession and the development of spruce silviculture. The composition of the vegetation of the ground stories is close to that of spruce forests (Formation A).

Formations C and D: pine-spruce forests, and pine forests with spruce and birch, locally with linden, oak and hazel. Pine and pine-spruce forests on uplands are not completely indigenous communities and represent a successional stage in transition to mature forest communities. The absence of pine regeneration in automorphic habitats indicates a derivative origin of pine forests after fires and fellings, as well as in the composition of silvicultural forests. In one case, succession is accompanied by active recovery of spruce forests (Formation C). In the other case, in habitats with nutrient-rich soils, the broad-leaved succession is observed (Formation D). The broad-leaved species displace the pine and pine-spruce communities after a few decades. Since the ecological range of the pine is wide, the pine forests can be found on soils with different texture and moisture regime. Respectively, the pine forests' typological diversity is higher compared to spruce forests (Table 1).

Formation E: oak forests with linden, spruce and birch. This group characterizes broad-leaved forests with a predominance of oak in the first story of the stand, the participation of spruce in the first and second sub-stories and species of the nemoral group in the lower stories of communities.

Formation F: broad-leaved-spruce forests. Forests with a mixed composition of spruce and broad-leaved species (oak, linden, maple) and species of the nemoral group in the lower stories of communities. This is primary (indigenous) communities, which are usually replaced by linden or spruce forests during felling.

Formation G: linden with oak, locally with spruce and birch. Broad-leaved forests with a predominance of linden in the first or second sub-stories and nemoral species in the lower stories. Spruce is occasionally represented in the upper stories and in undergrowth. Oxalis and some other boreal species are involved in the grass story. These communities are primarily only on the slopes of ravines and river valleys and on the uplands in the central and northern parts of the region. Here, they are derivatives of coniferous-deciduous forests [49,50]. Indigenous linden nemoral grass forests are represented only in the southern part of the region.

Formation H: birch forests with spruce and aspen, locally with oak and linden, and Formation I: aspen forests with spruce and oak. The predominance of small-leaved birch and aspen forests in the region is associated with the formation of young forests in fellings. In recent decades, it is also associated with a regenerative succession on massively abandoned tillage. Most of the community types are secondary and can develop in a wide range of habitat conditions. The typological diversity of small-leaved forest communities of birch and aspen is associated with a wide ecological tolerance and the ability to grow on soils of different texture, moisture and nutrient richness. However this forest formation creates conditions for the restoration of conditionally primary communities.

Formations J and K: gray and black alder. These communities are more often referred to as primary forests, preferring either waterlogged or rich brook habitats. However, there is another point of view, according to which gray alder forests are considered derivative forests, developing on the site of broad-leaved spruce communities [51,52] (Table 1). Each forest formation is represented by one or multiple association groups:

- Dwarf shrubs-small herb-green moss (DShG),
- Small herb (Sh),
- Small herb-broad herb (ShBh),
- Broad herb (Bh),
- Moist herb-broad herb (MhBh),
- Grass-marsh (Gm),

- Herb (H),
- Dwarf shrubs-herb-sphagnum (DHS).

Spatial rarefication for formations is performed in accordance with the methodology: A (10 km), B (10 km), C (1 km), D (5 km) and H (10 km). Two pairs of ecologically similar formations are merged: broadleaf formation (E and F) and alder formation (J and K). This made it possible to group the sample plots in accordance with the ecological similarity of syntaxa and achieve a relatively equal sample size: (82–112 sample plots in each sample).

The following non-forest habitats' field data were collected during field surveys: small leaf scrub (L), meadows (N), open marshy habitats (O) and agricultural fields (P). The following non-forest habitats are taken from the Global Forest Change dataset: cuts (M) and water objects (Q) [46]. Settlements (R) are taken from Openstreetmap (OSM) layers [53]. These habitats are mapped and merged with the final map through non-forest mask (Table 2).

**Table 2.** Number of non-forest field data points and data sources.

| Habitat Type | Small Leaf Scrub | Cuts | Meadows | Open Marshy Habitats | Agri Cultural Fields | Water Objects | Settlements |
|---|---|---|---|---|---|---|---|
| | L | M | N | O | P | Q | R |
| Number of points/source | 53 | Global forest watch (loss year) | 53 | 27 | 78 | Global forest watch (data mask) | Openstreetmap (OSM) |

### 3.2. Modeling of Formations

Ten different methods of grouping formations were tested (Table 3). In methods 1 and 2, all sample plots are used and each formation is modeled individually, and in method 2, the DEM is excluded from environmental variables. In methods 3–6, formations E, F and G (oak, linden, broadleaf forests) are combined, in method 3, the DEM is excluded from the environmental variables and in method 4, only spectral reflectances of July and September are left. Method 5 combines the H and J formations (birch and gray alder forests). Methods 7–9 apply spatial rarefication of occurrence data. In method 7, only the formation points A (spruce) are rarefied. In methods 8 and 9, different rarefication distances are applied for formations A, B, C, D and H. Two pairs of ecologically similar formations are merged: E with F and J with K. In method 9, the DEM is excluded from environmental variables.

**Table 3.** Modeling results for the 9 methods.

| Forest Plan Data | | Formations | Method of Modeling Proportion of Formation (%) | | | | | | | | | |
|---|---|---|---|---|---|---|---|---|---|---|---|---|
| | | | 1 | 2 | 3 | 4 | 5 | 6 | 7 | 8 | 9 | 8 [1] |
| Spruce | 24.4 | A | 3.2 | 3.3 | 7.0 | 7.7 | 5.25 | 5.0 | 12.6 | 7.6 | 6.6 | 7.0 |
| | | B | 4.8 | 13.7 | 11.2 | 15.0 | 9.30 | 19.2 | 13.2 | 16.0 | 15.1 | 18.1 |
| Pine | 20.7 | C | 5.4 | 4.6 | 7.5 | 4.5 | 2.84 | 5.8 | 4.4 | 2.5 | 2.5 | 2.5 |
| | | D | 9.7 | 9.8 | 15.4 | 10.0 | 17.39 | 11.6 | 8.9 | 15.5 | 14.4 | 16.0 |
| Oak | 1.7 | E | 14.0 | 9.7 | 11.0 | 10.3 | 11.72 | 10.5 | 9.9 | 4.1 | 9.0 | 2.9 |
| Broad leaf [2] | 0.08 | F | 17.5 | 19.4 | | | | | | | | |
| Linden | 0.64 | G | 0.7 | 1.7 | | | | | | 2.1 | 2.0 | 4.4 |
| Birch | 39.6 | H | 16.0 | 14.3 | 35.9 | 31.2 | 15.38 | 22.7 [3] | 30.0 | 32.5 | 35.5 | 31.0 |
| Aspen | 8.4 | I | 2.6 | 9.3 | 4.9 | 6.6 | 21.23 | 12.6 | 6.4 | 6.2 | 6.1 | 5.1 |
| Grey alder | 2.3 | J | 15.3 | 4.4 | 7.1 | 14.8 | 16.88 | - | 14.7 | 13.5 | 9.0 | 13.0 |
| Black alder | 1.8 | K | 10.6 | 9.9 | | | | 12.6 | | | | |

[1] Method 8 including Palsar-2 dataset. [2] Broadleaf in Forest Plan includes maple, ash and elm. [3] Birch including gray alder.

The results of the eighth method of grouping of formations are closest to the ratio of tree species given in the Forest Plan (FP) [17]. The modeling is re-run with settings of the eighth method and addition of the Palsar-2 dataset (Figure 1). The results of modeling the dominant formations are in good agreement with the data of the Forest Plan: birch 30.1% (39.6% in the FP), spruce 25.1% (24.4% FP), pine 18.5% (20.7% FP), aspen 5.1% (8.4%) and oak with broad-leaved (E + F) 2.9% (1.78% FP). The proportions of linden (G) and alder (J + K) are overestimated by 7.3 and 3.1 times respectively, compared to the data of the Forest Plan (Figure 2). Several possible reasons for this discrepancy are suggested. First, there is a lack of field data. Naturally, in the absence of preliminary stratification, such formations will rarely be encountered during field routes. At the same time, despite the fact that the number of points of the rare formations has been conditionally brought to the level of 80–100, they nevertheless still have a significant role of autocorrelation: often these points are located in spatial clusters, which reduces the quality of models. However, another factor is also important. In the course of field work, it was repeatedly noticed that the forests indicated on the forest inventory maps as birch are in fact alder forests.

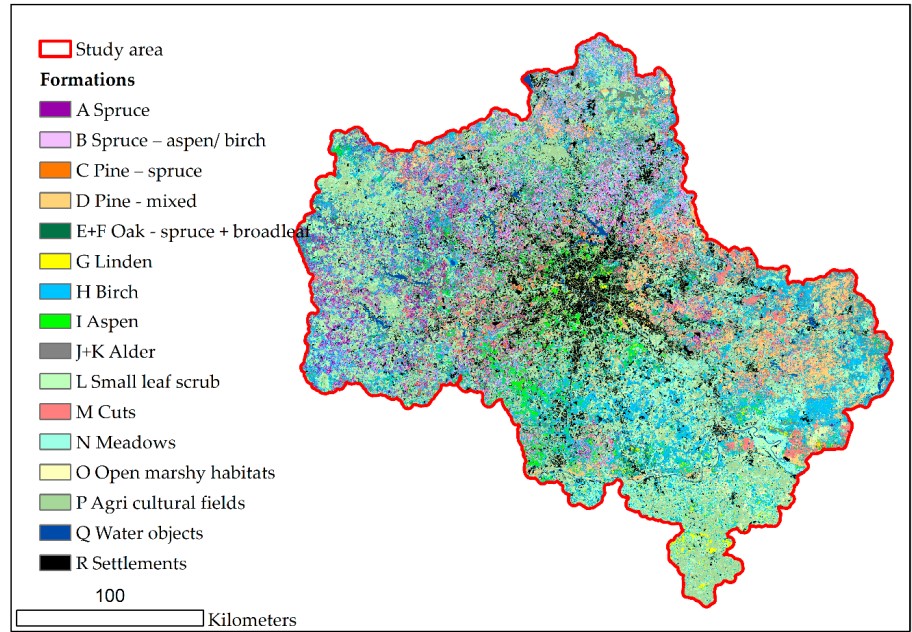

**Figure 2.** Cartographic model of forest formations and non-forest habitats.

According to the confusion matrix (Table 4), the classification accuracy is 0.46. The best classification quality is for alder forests. This is noteworthy in the context of the above difference with the official data of the Forest Plan. A low level of matching with field data is found for the formations of aspen (I) 0.31, spruce forests (A) 0.33 and oak broad-leaved forests (E + F) 0.34. Aspen forests (I) are poorly separated from birch (H) forests. For spruce forests (A), matching problems are associated with a closely related formation: spruce-small-leaved forests (B). Oak broad-leaved forests (E + F) are poorly separated from spruce-small-leaved (B) and birch (H) forests. The use of the Palsar-2 dataset allows to increase overall accuracy from 0.44 to 0.45, and it gives a rather significant accuracy increase for pine (D), 0.57 to 0.62, and linden (H), 0.35 to 0.45 (confusion matrix without the radar dataset is not demonstrated here).

**Table 4.** Confusion matrix model of forest formations.

| Formation | A | B | C | D | E + F | G | H | I | J + K | Total | User Accuracy |
|---|---|---|---|---|---|---|---|---|---|---|---|
| A | 30 | 8 | 15 | 8 | 10 | 4 | 4 | 0 | 3 | 82 | 0.37 |
| B | 25 | 39 | 8 | 5 | 11 | 4 | 11 | 15 | 7 | 125 | 0.31 |
| C | 13 | 1 | 36 | 7 | 0 | 0 | 2 | 0 | 0 | 59 | 0.61 |
| D | 4 | 9 | 24 | 51 | 0 | 2 | 6 | 4 | 3 | 103 | 0.50 |
| E + F | 2 | 4 | 1 | 0 | 29 | 8 | 4 | 12 | 2 | 62 | 0.47 |
| G | 0 | 1 | 0 | 1 | 2 | 46 | 5 | 5 | 3 | 63 | 0.73 |
| H | 13 | 18 | 4 | 9 | 21 | 17 | 51 | 23 | 4 | 160 | 0.32 |
| I | 3 | 6 | 0 | 1 | 8 | 16 | 7 | 28 | 3 | 72 | 0.39 |
| J + K | 2 | 0 | 0 | 0 | 5 | 5 | 4 | 4 | 54 | 74 | 0.73 |
| Total | 92 | 86 | 88 | 82 | 86 | 102 | 94 | 91 | 79 | 800 | 0.37 |
| P_Accuracy | 0.33 | 0.45 | 0.41 | 0.62 | 0.34 | 0.45 | 0.54 | 0.31 | 0.68 | | 0.46 |
| Kappa | | | | | | | | | | | 0.39 |

### 3.3. Modeling of Association Groups

Despite the uneven spatial distribution of points, modeling of association groups is performed without spatial rarefication. Table 5 shows the results of modeling of association groups. The average level of points matching between all association groups is 0.29 and it ranges from 0.03 to 0.69. On average, a low percentage of points matching is typical for coniferous groups of associations (A–D) and broad-leaved conifers (F). Oak-spruce (E), linden (G), birch (H) and aspen (I) have an average level of recognition. The best quality is for gray alder (J) and black alder (K) formations.

According to the assessment of the spatial distribution of the identified association groups of forest communities (Table 5), the largest area (12.5%) is occupied by communities of derivative birch and aspen forests (Formations H and I) with a predominance of the mesotrophic and hydromorphic series (Bh, MhBh, Gm). The proportion (4.7%) of birch (H) grass-marsh forests (Gm), which are distinguished by strong recreational disturbance, is also high. Within the pine formation (D), the maximum proportion (about 5%) is occupied by communities (DShG and Sh), which tend to succession towards boreal spruce forests. The same pattern is observed in the formation of mixed spruce-small-leaved forests (B), occupying an area (4.72%) where their successional dynamics are also directed towards the restoration of spruce communities of groups (DShG and Sh). Another part of these communities of formation B (3.3%), in terms of the composition of the ground layer (ShBh and Bh), has a tendency to succession towards broad-leaved communities with a nemoral composition of the ground layer. It is obvious that this group is of artificial origin. The area of communities within the spruce forests (A) is small and varies in the range of 0.47–1.37%.

Table 6 shows the proportion of the area occupied by non-forest habitats.

**Table 5.** Percentage of total area by association groups (upper number) and proportion of matching points (lower number).

| Formations | Spruce | Spruce-Aspen/Birch | Pine-Spruce | Pine | Oak-Spruce | Broad-Leaf Spruce | Linden | Birch | Aspen | Grey Alder | Black Alder |
|---|---|---|---|---|---|---|---|---|---|---|---|
| Association groups | A | B | C | D | E | F | G | H | I | J | K |
| DShG | $\frac{0.47}{0.21}$ | $\frac{3.07}{0.27}$ | $\frac{0.36}{0.21}$ | $\frac{2.44}{0.43}$ | | | | | | | |
| Sh | $\frac{1.37}{0.24}$ | $\frac{1.65}{0.14}$ | $\frac{0.41}{0.2}$ | $\frac{2.51}{0.14}$ | | | | $\frac{0.28}{0.33}$ | | | |
| ShBh | $\frac{0.62}{0.11}$ | $\frac{1.58}{0.14}$ | $\frac{0.38}{0.33}$ | $\frac{0.19}{0.03}$ | | | | $\frac{1.18}{0.08}$ | | | |
| Bh | $\frac{1.31}{0.2}$ | $\frac{1.71}{0.18}$ | $\frac{0.08}{0.19}$ | $\frac{1.12}{0.38}$ | $\frac{1.10}{0.38}$ | $\frac{1.01}{0.19}$ | $\frac{1.73}{0.35}$ | $\frac{2.85}{0.25}$ | $\frac{1.73}{0.28}$ | | |
| MhBh | | | | | | | | $\frac{1.89}{0.06}$ | $\frac{1.36}{0.5}$ | $\frac{1.44}{0.69}$ | $\frac{0.46}{0.43}$ |
| Gm | | | | | | | | $\frac{4.70}{0.35}$ | | | $\frac{4.55}{0.57}$ |
| H | | | | $\frac{0.10}{0.23}$ | | | | $\frac{3.72}{0.56}$ | | | |
| DHS | | | | $\frac{0.91}{0.55}$ | | | | $\frac{1.32}{0.5}$ | | | |
| Mean % of point matching | 0.19 | 0.18 | 0.23 | 0.29 | 0.38 | 0.19 | 0.35 | 0.3 | 0.39 | 0.69 | 0.5 |

Italic: level of points matching.

**Table 6.** Percentage of total area by non-forest land cover types.

| Habitat Type | Small Leaf Scrub | Cuts | Meadows | Open Marshy Habitats | Agri Cultural Fields | Water Objects | Settlements |
|---|---|---|---|---|---|---|---|
| | L | M | N | O | P | Q | R |
| % total cover | 14.69 | 4.16 | 7.07 | 2.34 | 12.85 | 1.08 | 9.23 |

## 4. Discussion

A wide range of statistical models exist to model the spatial structure of natural features and phenomena: Resource Selection Function, Generalized Linear Models, Artificial Neural Networks, Maximum Entropy and Classification and Regression Trees [21]. Formerly for the Moscow Region, it was shown that Linear Discriminant Analysis demonstrates satisfactory results for modeling the spatial structure of forests based on field data and Landsat 8 spectral reflectance supplemented with digital elevation model and their derivative parameters [22]. However, linear functions of multispectral data demonstrate limited capabilities, which is demonstrated in our study. The average proportion of matching between field data and model was 52%. It varied from 20% to 100% for 38 association groups based on 1025 field sample plots. However, the number of sample plots per each association group varied from 4 to 114, which is unlikely in terms of sample size [54].

Previous works have shown the advantages of ecological-phytocenotic classification over ecological-floristic for the purpose of mapping. The relative quality of the discriminant analysis of the identified syntaxa within the ecological-floristic classification demonstrated a lower accuracy of the ecological-floristic classification (69.7%) compared to the ecological-phytocenotic classification (78.6%) [55].

The assessment of the spatial diversity of forest cover is connected with a number of limitations. Within the current study, the limitations may be generalized into three groups. The group of natural factors makes the most irregular and heterogeneous input into classification uncertainty. Foremost, the presence of multidirectional processes with degression dynamics (recreational impact, road and construction infrastructure) and restoration dynamics (tillage abandoning, forest silviculture). This limitation significantly disrupts the natural composition of coenotic types. The study area is located in the zone of Eastern European deciduous-coniferous forests, characterized by a mixed polydominant composition, which is difficult to analyze the species composition of communities and their classification, which has also been noted by other authors [56,57]. With regard to deciduous forests, an increase in their proportion due to climate warming, as well as the warming effect of the megalopolis, cannot be ruled out [58]. The polydominance of the tree layer [49], transitional succession status of most derivative forests [59], anthropogenic disturbance, as well as a high proportion of forest silviculture in the region [60], together with the anthropogenic impact mentioned above, make it one of the most complicated regional study areas.

Another group of classification uncertainty factors is generated by the quality and properties of environmental variables, especially Landsat spectral reflectance. Even excluding uncertainties of nadir angle, radiometric correction and atmospheric transparency, each of them varies by 5% to 7.5% [61], and there is still one important issue: the area of the Landsat pixel (0.09 hectares) slightly exceeds the area of the sample plot (0.04 hectares). In terms of statistics, it means that within each sample plot, one cannot evaluate statistical parameters of each sample of pixels and reflectance data are used with all potential extremes and outbreaks. Simply speaking, the Landsat dataset is too coarse for modeling typical $20 \times 20$ m sample plots [62]. A few solutions might be discussed in this context. One obvious solution is use of higher resolution imagery—Sentinel-2. Not to be overlooked is filtering of spectral reflectance, i.e., median filter [63].

The third group of uncertainty factors is considered in the Introduction Section, and it concerns the uneven spatial distribution of field data, typical for the regions of the Russian Federation, which is a critical factor affecting the quality of the models. This group might be characterized as of human and organizational origin.

The method applied in the current study made it possible to obtain a map of the spatial structure of the formations corresponding to the official data for most of the species (birch, spruce, pine, oak, broad-leaved species and linden) and slightly overestimated results for some rarely distributed species (alder and aspen). The applied set of methods allowed to reach overall accuracy of 0.46 for forest formations.

Modeling more detailed syntaxa of forest cover, association groups, for which spatial rarefication of points is not used, emphasizes the negative contribution of uneven field data. This is especially important for coniferous and broad-leaved coniferous formations. However, the spatial pattern is quite plausible, the results are consistent with the previously developed forest cover models for part of the Moscow Region based on discriminant analysis [43,44], and also with a map of the vegetation cover of the Moscow Region [18].

An attempt at large-scale mapping is promising for assessing biodiversity and forest dynamics, but it has limitations in the area of study with the characteristic physical and geographical diversity of the territory. The overall technology holds promise, but still, the uncertainty of classification is rather low and one shall look for utilizing the higher spatial resolution datasets along with filtering approaches. The previously performed work in the southwestern part of the Moscow Region demonstrated higher quality of the cartographic model (78.6%) of the distribution of 15 types of forest communities [55]. Thus, for large regions with a complex natural structure and anthropogenic history, it might be useful to perform modeling within the individual landscape structures.

The results obtained underline the need to use the resulting map as a stratification matrix and to carry out additional field research, systematic and optimally justified. Additional field research should be aimed at achieving the following objectives:

- Creation of the set of field descriptions, evenly distributed in space and taking into account rare and remote habitats.
- Bringing the minimum number of descriptions of association groups to at least 50 (additional 494 descriptions), and in the long term, to 80 (1240 additional descriptions).

## 5. Conclusions

The use of MaxEnt nonlinear modeling together with additional tools (geographically structured spatial jack-knifing, spatial rarefication of occurrence data and independent testing of model feature classes and regularization parameters) can be used to manage the problem of uneven distribution of field data and to attempt to create a probabilistic cartographic model of forest formations at the regional level. The results of our modeling correspond well to the official data of forest inventory despite the high level of modeling uncertainty.

The main limitations of identifying and assessing the spatial distribution of types of forest communities at a more detailed typological level, in the rank of groups of associations, additionally to those mentioned above, including a series of studies at the sub-regional level within territorial units of natural zoning, were formulated. The need to utilize higher spatial resolution datasets along with filtering was emphasized.

The resulting cartographic model of the groups of associations can be used to stratify the study area and plan the optimal number and placement of field routes necessary for the final statistically valid model of forest communities.

**Author Contributions:** I.K. designed the study and performed the modeling; I.K. and T.C. did research and the data-analysis; T.C. performed ecological-phytocenotic classification of forest communities. All authors have read and agreed to the published version of the manuscript.

**Funding:** The Russian Science Foundation (project no. 18-17-00129) supported this study. This study is also conducted in the framework of the Institute of Geography RAS (project no. 0148-2019-0007) in terms of studying the composition of forest communities.

**Acknowledgments:** The authors thank all colleagues for participating in field surveys and discussing the modeling design: Olga Morozova, Elena Suslova, Nadejda Beliaeva and Maria Arkhipova.

**Conflicts of Interest:** The authors declare no conflict of interest.

## Appendix A

**Table A1.** List of initial environmental variables, filtered through autocorrelation analysis.

| # | Sensor | Mosaic Date | Index | Removed Due to Autocorrelation > 95% |
|---|--------|-------------|-------|--------------------------------------|
| 1 | SRTM | 2009 | Elevation (meters) | |
| 2 | SRTM | 2009 | Slope (degrees) | |
| 3 | SRTM | 2009 | Aspect | |
| 4 | SRTM | 2009 | Shaded relief | |
| 5 | SRTM | 2009 | Profile Curvature | |
| 6 | SRTM | 2009 | Plan Convexity | |
| 7 | SRTM | 2009 | Longitude Convexity | yes |
| 8 | SRTM | 2009 | Cross Sectional Convexity | |
| 9 | SRTM | 2009 | Minimum Curvature | |
| 10 | SRTM | 2009 | Maximum Curvature | |
| 11 | SRTM | 2009 | Elevation Root Mean Square Error | |
| 12 | SRTM | 2009 | Slope (percent) | yes |
| 13 | SRTM | 2009 | Laplacian | |
| 14 | Landsat 8 | March 2019 | Band 1 | |
| 15 | Landsat 8 | March 2019 | Band 2 | yes |
| 16 | Landsat 8 | March 2019 | Band 3 | yes |
| 17 | Landsat 8 | March 2019 | Band 4 | yes |
| 18 | Landsat 8 | March 2019 | Band 5 | |
| 19 | Landsat 8 | March 2019 | Band 6 | |
| 20 | Landsat 8 | March 2019 | Band 7 | yes |
| 21 | Landsat 8 | March 2019 | EVI | |
| 22 | Landsat 8 | March 2019 | MSAVI | |
| 23 | Landsat 8 | March 2019 | NBR | |
| 24 | Landsat 8 | March 2019 | NBR2 | yes |
| 25 | Landsat 8 | March 2019 | NDMI | yes |
| 26 | Landsat 8 | March 2019 | NDVI | yes |
| 27 | Landsat 8 | March 2019 | SAVI | |
| 28 | Landsat 8 | May 2019 | Band 1 | |
| 29 | Landsat 8 | May 2019 | Band 2 | yes |
| 30 | Landsat 8 | May 2019 | Band 3 | yes |
| 31 | Landsat 8 | May 2019 | Band 4 | yes |
| 32 | Landsat 8 | May 2019 | Band 5 | |
| 33 | Landsat 8 | May 2019 | Band 6 | |

**Table A1.** *Cont.*

| # | Sensor | Mosaic Date | Index | Removed Due to Autocorrelation > 95% |
|---|--------|-------------|-------|--------------------------------------|
| 34 | Landsat 8 | May 2019 | Band 7 | yes |
| 35 | Landsat 8 | May 2019 | EVI | |
| 36 | Landsat 8 | May 2019 | MSAVI | |
| 37 | Landsat 8 | May 2019 | NBR | |
| 38 | Landsat 8 | May 2019 | NBR2 | yes |
| 39 | Landsat 8 | May 2019 | NDMI | yes |
| 40 | Landsat 8 | May 2019 | NDVI | |
| 41 | Landsat 8 | May 2019 | SAVI | |
| 42 | Landsat 8 | July 2019 | Band 1 | |
| 43 | Landsat 8 | July 2019 | Band 2 | |
| 44 | Landsat 8 | July 2019 | Band 3 | |
| 45 | Landsat 8 | July 2019 | Band 4 | |
| 46 | Landsat 8 | July 2019 | Band 5 | |
| 47 | Landsat 8 | July 2019 | Band 6 | |
| 48 | Landsat 8 | July 2019 | Band 7 | |
| 49 | Landsat 8 | July 2019 | EVI | |
| 50 | Landsat 8 | July 2019 | MSAVI | |
| 51 | Landsat 8 | July 2019 | NBR | |
| 52 | Landsat 8 | July 2019 | NBR2 | yes |
| 53 | Landsat 8 | July 2019 | NDMI | yes |
| 54 | Landsat 8 | July 2019 | NDVI | |
| 55 | Landsat 8 | July 2019 | SAVI | |
| 56 | Landsat 5 | July 2010 | Band 1 | |
| 57 | Landsat 5 | July 2010 | Band 2 | yes |
| 58 | Landsat 5 | July 2010 | Band 3 | yes |
| 59 | Landsat 5 | July 2010 | Band 4 | yes |
| 60 | Landsat 5 | July 2010 | Band 5 | |
| 61 | Landsat 5 | July 2010 | Band 6 | |
| 62 | Landsat 5 | July 2010 | Band 7 | yes |
| 63 | Landsat 5 | July 2010 | EVI | |
| 64 | Landsat 5 | July 2010 | MSAVI | |
| 65 | Landsat 5 | July 2010 | NBR | |
| 66 | Landsat 5 | July 2010 | NBR2 | yes |
| 67 | Landsat 5 | July 2010 | NDMI | yes |
| 68 | Landsat 5 | July 2010 | NDVI | yes |
| 69 | Landsat 5 | July 2010 | SAVI | |
| 70 | Landsat 5 | September 2019 | Band 1 | |
| 71 | Landsat 8 | September 2019 | Band 2 | yes |
| 72 | Landsat 8 | September 2019 | Band 3 | yes |
| 73 | Landsat 8 | September 2019 | Band 4 | yes |

**Table A1.** *Cont.*

| # | Sensor | Mosaic Date | Index | Removed Due to Autocorrelation > 95% |
|---|--------|-------------|-------|--------------------------------------|
| 74 | Landsat 8 | September 2019 | Band 5 | |
| 75 | Landsat 8 | September 2019 | Band 6 | |
| 76 | Landsat 8 | September 2019 | Band 7 | yes |
| 77 | Landsat 8 | September 2019 | EVI | |
| 78 | Landsat 8 | September 2019 | MSAVI | |
| 79 | Landsat 8 | September 2019 | NBR | |
| 80 | Landsat 8 | September 2019 | NBR2 | yes |
| 81 | Landsat 8 | September 2019 | NDMI | |
| 82 | Landsat 8 | September 2019 | NDVI | |
| 83 | Landsat 8 | September 2019 | SAVI | |
| 84 | Palsar-2 | 2019 | HH polarization | |
| 85 | Palsar-2 | 2019 | HV polarization | |

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
