# Peer review of "Modeling of Forest Communities’ Spatial Structure at the Regional Level through Remote Sensing and Field Sampling: Constraints and Solutions"

_forests, doi:10.3390/f11101088_

Round 1

Reviewer 1 Report

The study presented in this manuscript attempts to provide a predictive modelling approach for forest types for a region in which field observations are of limited scope and quality. Some variables are iterated upon in the modelling process, such as the scale of spatial scale of aggregation of plots, and the combination of groups.

I am sympathetic to limitations to the study design arising from the limits of the available data, since this is the motivation for the study in the first place. Inferential statistical analyses, for example, would be inappropriate due to problems of data-snooping and multiple comparisons. However, the discussion is very light, especially discounting the background information that comprises half of its content. Additionally, the manuscript itself is held back by its communication, structure, and polish. There are several recurring problems with the communication:

1) Improperly structured lists (many lists are missing "and" to precede the last element).

2) Many abbreviations are not defined when they are first used.

3) "Maxent" and "Moscow region" change in their capitalization pattern throughout. There are other similar inconsistencies in the presentation of terminology.

4) There is a consistent use of hyphens and en-dashes where colons would be more appropriate.

5) The perspective and tense switches many times.

6) Some of the material is distributed oddly between the sections. For example, there are several avenues of discussion explored in the Results, and lines 339-360 in the Discussion are the sort of context for the study that I would expect to find in the Introduction.

Additionally, here are some minor, but abundant, corrections and suggestions:

Line 56: Remove period after “samples”.

Line 62: “carried out” does not convey a spatial meaning, but rather implies a temporal meaning.

Line 76 – 77: Consider revising “the importance… is extremely important”

Line 78: “NGO” is not previously established as an abbreviation.

Line 80 and line 82: Amend capitalization of “Region”. It is not capitalized elsewhere.

Line 88: “first of all” does not seem to fit here. Perhaps “primarily” might convey the intended meaning?

Line 90: Consider ending sentence after “[20]”.

Line 91: “is” to “was”.

Line 92: “In current” to “In the current”.

Line 93: “maxent” to “MaxEnt” as per the next line?

Line 95: “and robust” to “and is robust”?

Line 96: “Maxent” to “MaxEnt”? Change all instances to the same capitalization pattern.

Line 97: “For instance” cannot start the sentence unless the subject from the previous sentence is re-established. Consider adjoining to the previous sentence with a comma.

Line 98: “STH” is not previously established as an abbreviation.

Line 98-99: The list does not correctly terminate with an “and” or similar.

Line 100: “as model” to “as a model” or “as the model”.

Line 103: “as a” to “are a”? Or a similar correction to work with “We can assume”.

Line 105: “to spatial modeling” to “spatially model” or “to facilitate spatial modeling” or similar.

Line 106: “on the example” to “for the example”

Line 110: Remove “including”.

Line 119: “spruce forests natural range” to “natural range of the spruce forests” or “spruce forests’ natural range”.

Line 122: “WW2” is not established as an abbreviation.

Line 126: “energy complex,” to “energy complexes and”

Line 129-130: This sentence is structurally incomplete. Also, the references should be combined into a single set of brackets.

Line 132-134: This sentence is structurally incomplete.

Figure 1 caption: Are the “Formations” the relevés, or a set thereof? This is not explained until line 159. Provide more information in the caption.

Line 137: “incl.” to “including”.

Line 142: “According to methodology” to “According to the methodology”.

Lines 147-152: Capitalize the first word of bullet points that contain complete sentences. Additionally, this list has clearly been formed by breaking up a paragraph of prose, and therefore the internal structure needs amending (for example, the third bullet point reads as a statement of action, rather than as a property recorded at the relevé).

Line 159: “presented” to “represented”?

Line 160, 162: “storeys” to “stories” if you want to follow US English conventions.

Line 170: Extraneous space before period.

Line 170-171: Several erroneous spaces, and also a symbol-code appearing in the text in the review version.

Line 173: “anb” to “and”, and an erroneous space before the period.

Line 173: “This called” to “This is called”.

Line 177: “allows to use the” needs amending. Perhaps “allows this study to use the”?

Line 184: Was “SRTM” previous established as an abbreviation?

Line 186: “with” to “the”? and “which” to “that”?

Line 190: “which” to “that”.

Line 191: Homogeneous is used elsewhere in the manuscript, instead of homogenic.

Line 192-193: “divisional,” to “divisional, in that” and “more sensitive natural object” to “a more sensitive natural object” or “more sensitive natural objects”.

Line 196: “stratum” to “strata”.

Line 197: “forest strata” to “a forest stratum”, and “non-forest strata” to “a non-forest stratum”.

Line 197: “stratum” to “strata”!

Line 199: “AUC” is not PEAAA!

Line 201: “use systematic” to “use a systematic” and “which” to “, which”

Line 204: Somewhere in “allows all to use all feature class types only over” the meaning of the sentence is lost. Please revise.

Line 210: “by method” to “by the method”

Line 214: “of number” to “of the number”

Line 218: “The Table 1” to “Table 1”

Line 222: “pine, locally” to “pine, and locally”? Depending on the intended meaning.

Lines 222-235: Table 1 communicates the information in this section more effectively, unless there is any particularly notable feature that I am missing. Additionally, the abbreviations used for the association groups need to be established in Table 1’s caption.

Line 238: “This made possible” to “This made it possible”

Line 246: “data was” to “data were”

Line 249: “with final map” to “with the final map”

Line 270: “will rarely come across” to “will rarely be encountered”? Or a similar amendment.

Line 279: “the alder” to “alder”

Lines 288-289: The sentence beginning “Low level” seems to be lacking a subject.

Lines 299-300: Can an “average” be “typical”?

Table 5: Is the first number in each cell the percentage, and the second the proportion? I would guess so, but it is unclear, especially with the asterisk (with no corresponding partner) that states the italicized numbers are a “level”.

Line 314: “dynamics is” to “dynamics are”

Line 379: “with a” to “with the”

Lines 388-391: These bullet points clearly used to be a list. Please amend the capitalization and punctuation appropriately.

Author Response

We are very thankful to Reviewer. A lot of style, syntaxis and grammar corrections made according to Reviewer's notes.
We also made  some restructurization of material between manuscript sections and tried to improve Discussion and Conclusions.

Reviewer 2 Report

Weak correlations and results need beefing up in discussion and some more fully fleshed out conclusions. suggestions are in the comments.

Landsat and SRTM-DEM data is a bit coarse for 20 by 20 M study plots and is it not surprising that the was had low correspondence to the classification this needs to be discussed in the discussion and in the conclusions

Author Response

We are very thankful to Reviewer. We agree with all notes and corrections and update the ms according to them. However in some sections some significant amendments can be seen according to other Reviewer's comments.

Round 2

Reviewer 1 Report

Dear authors,

Thank you very much for the detailed response and attention to my comments. The quality of the science in the manuscript is now apparent with the much-improved quality of the communication and presentation. An interesting piece of work.